# Site-Specific Fluorescent Labeling of RNA Interior Positions

**DOI:** 10.3390/molecules26051341

**Published:** 2021-03-03

**Authors:** Barry S. Cooperman

**Affiliations:** Department of Chemistry, University of Pennsylvania, Philadelphia, PA 19104, USA; cooprman@pobox.upenn.edu; Tel.: +12-15-898-6330

**Keywords:** site-specific fluorescent labeling, RNA interior positions

## Abstract

The introduction of fluorophores into RNA for both in vitro and in cellulo studies of RNA function and cellular distribution is a subject of great current interest. Here I briefly review methods, some well-established and others newly developed, which have been successfully exploited to site-specifically fluorescently label interior positions of RNAs, as a guide to investigators seeking to apply this approach to their studies. Most of these methods can be applied directly to intact RNAs, including (1) the exploitation of natural posttranslational modifications, (2) the repurposing of enzymatic transferase reactions, and (3) the nucleic acid-assisted labeling of intact RNAs. In addition, several methods are described in which specifically labeled RNAs are prepared de novo.

## 1. Introduction

The introduction of fluorophores into RNA for both in vitro and in cellulo studies of RNA function and cellular distribution is a subject of great current interest. Excellent comprehensive reviews have recently been published on the synthesis of fluorescent analogs of nucleosides [1,2,3,4,5], and on the introduction of fluorescent probes into RNA by chemical and/or enzymatic procedures [6,7,8,9], which include descriptions of the utilization of changes in fluorescence to elucidate biological, biochemical and biophysical mechanisms. Against this background, my goal in writing this chapter is to briefly review methods, some well-established and some newly developed, that have been successfully exploited to site-specifically fluorescently label interior positions of RNAs, as a guide to investigators seeking to apply this approach to their studies. 

## 2. Exploitation of Natural Posttranslational Modifications

There are more than 100 known posttranscriptional modifications of tRNAs [10]. Of these, four are pertinent for this review. Wybutosine is fluorescent, and dihydrouridine (DHU), 3-amino-3-carboxypropyluridine, acp^3^U, and 4-thiouridine, s^4^U can be derivatized with appropriately reactive fluorophores, as described below.

### 2.1. Wybutosine

The fluorescent nucleoside wybutosine, or its derivative hydroxy-wybutosine (Figure 1), is found uniquely at position 37 of eukaryotic tRNA^Phe^s [11,12]. Wybutosine fluorescence in yeast tRNA^Phe^ is environmentally sensitive [13,14]. This property has been exploited in measuring yeast tRNA^Phe^ binding to various ligands [15,16] and in elucidating reaction mechanisms during ribosome-catalyzed polypeptide synthesis [17,18]. However, wybutosine’s comparatively low extinction coefficient (ε_310–315_ ~ 4000) [19,20] and low quantum yield (0.07) [13] limit its general utility as a fluorescence probe. 

### 2.2. DHU

DHU is formed by post-transcriptional reduction of uridine and is the most widespread modified base which can be straightforwardly labeled with fluorescent groups. Four dihydrouridine synthases (Dus1–Dus4) have been identified, which use NADPH as the external reductant that is coupled with an enzyme-bound FMN cofactor. Collectively the Dus enzymes are responsible for the presence of one or more DHUs in the d-loop of the overwhelming majority of tRNAs, located at positions 16, 17, 17a 20, or 20a [21], as well as the much rarer instances of DHU within the variable loop of some tRNA^Tyr^s [21]. One or more of the Dus enzymes appear to be responsible for most, if not all, of the DHUs recently found to be present at 143 sites within 125 fission yeast mRNAs [22]. A practical consequence is that DHU can be installed as a potential site for introduction of fluorophores not only into unmodified tRNA transcripts [23,24], but also for a wider variety of sites within RNAs in general.

Zachau and Wintermeyer and their co-workers demonstrated that NaBH_4_ reduction in DHU residues allowed covalent introduction of weak base NH_2_-containing fluorophore nucleophiles into tRNAs with substantial retention by the labeled tRNAs of protein synthesis activity. Their work made extensive use of proflavin derivatives of tRNA [25]. More recently rhodamine 110 [22,23] and hydrazide derivatives of both Cy dyes [26,27] and Alexa dyes [28] have been incorporated into tRNAs. Despite this rather extensive use of fluorescent tRNAs labeled at DHU positions, the precise chemistry of the reaction leading to fluorophore introduction into these positions was for a long time unclear. This situation was remedied by the demonstration that the labeling reaction proceeded via formation of tetrahydrouridine (THU), as shown in Figure 2 [29]. The second step in this scheme likely proceeds via a dehydration intermediate that is attacked by the weak base amine, in an overall general acid-catalyzed Michael addition.

tRNAs fluorescently labeled via DHU chemistry have been used widely in in vitro ensemble [30,31,32] and single-molecule [33,34,35,36] studies of protein synthesis reaction mechanisms that exploit changes in fluorescence spectra, quantum yield, fluorescence anisotropy and/or FRET efficiencies. They have also found increasing use in intact cell studies, into which they are introduced via transfection [37], electroporation [38], microinjection [39] or rapid, pressure-induced generation of transient pores in cell membranes [28]. A common application has been for monitoring rates of protein synthesis. This approach is based on measuring the FRET intensities that are generated when a donor-labeled tRNA binds next to an acceptor-labeled tRNA in adjacent sites on an active ribosome. When bulk tRNA labeled with both donor and acceptor dyes is employed, the FRET signal provides a measure of overall protein synthesis [28,37,40,41,42,43]. In contrast, FRET signals that arise from ribosome-bound isoacceptor tRNA pairs which are cognate to specific mRNA dicodons measure synthesis of specific proteins [43,44,45]. Monitoring the movement of tRNAs within cells is a second application, and has been employed to measure tRNA diffusion rates in bacteria [38,46], tRNA trafficking between cytoplasm and nucleus in mouse embryonic fibroblasts [39], and dynamic mixing of tRNAs between neurite granules and rapid, bi-directional vectorial movement within neurites [41].

### 2.3. acp^3^U

A recently identified enzyme, tRNA U47 acp transferase A, abbreviated as TuaA, is responsible for modifying eight *E. coli* tRNAs, each at position 47 within the variable loop, by transfer of acp from S-adenosylmethionine (SAM) [47]. Two other enzymes also transfer acp from SAM to RNA: Tyw2/Trm12, which does so as an intermediate step in the synthesis of the tRNA wybutosine modification [48], and Tsr3 as part of the hypermodification of the yeast 18S rRNA U1191 to m^1^acp^3^ Ψ [49]. The amine group in acp is readily modified by amine-specific reagents, e.g., *N*-hydroxysuccinimide esters of Cy dyes [50]. The resulting fluorescent tRNAs have been used in single molecule studies of the mechanism of ribosome-catalyzed protein synthesis [51,52,53,54,55].

### 2.4. s^4^U

The tRNA modification enzyme ThiI, recently renamed tRNA thiouridine I, (TtuI) [56] catalyzes the ATP-dependent formation of 4-thiouridine, s^4^U, in bacteria and archaea, using cysteine as the ultimate sulfur donor, via a complex set of reactions. The s^4^U nucleoside is almost always found at position 8 of most bacterial and archaeal tRNAs. As TtuI modifies U8 of all bacterial tRNAs, there is no RNA sequence motif determining the specificity for U8 thiolation. Lauhon et al. [57] determined that a truncated tRNA consisting of 39 nucleotides (TPHE39A) mimicking the 3′-terminus, acceptor stem and T-stem loop of *E. coli* tRNA could serve as a minimal substrate for s^4^U8 synthesis (Figure 3). A subsequent structure of the TtuI-TPHE39A structure shows the enzyme recognizes the highly conserved 3′-ACCA end of tRNA and acts as a molecular ruler defining the length from the 3′-ACCA end to the site of modification [58], but successful efforts to further reduce substrate size have not yet been reported.

Cy-maleimides have been used to derivatize s^4^U, and the resulting fl-tRNAs have been used in single molecule studies of protein synthesis [50]. However, the yield of the derivatization reaction is quite low, due to the relatively poor reactivity of the 4-thione group toward maleimide. A more recent strategy for labeling s^4^U at much higher efficiency with exquisite specificity for s^4^U utilizes a two-step scheme in which the thione is first oxidized by periodate to the sulfonate, which is subsequently displaced by fluoresceinamine (FAM-NH_2_) [59].

### 2.5. Extensions beyond tRNA Labeling

It has generally been assumed that fluorescent labeling via DHU, acp^3^U, and s^4^U chemistry would be restricted to tRNAs. The recent demonstration that DHU is also found in many mRNA sites [22] raises the possibility that fluorescent labeling via DHU chemistry could be extended to other RNA targets. The potential of this approach will become clearer as the preferred secondary structure [60] and/or sequence contexts for DHU formation are defined. It is interesting to consider whether these recent results showing DHU presence in mRNAs will spur investigations not only into whether acp^3^U and s^4^U modifications are also present in RNAs other than tRNA, but also whether directed evolution of the enzymes involved in DHU, acp^3^U, and s^4^U formation would be successful in broadening substrate specificities [61].

## 3. Repurposing Enzymatic Transferase Reactions

Two recent reviews provide excellent overviews of the repurposing of enzymes to site-specifically label RNAs, at internal positions as well as at the 5′-mRNA cap and 3′-terminus [7,8]. Here we briefly present some points particularly pertinent to site-specific internal RNA position fluorescent labeling and highlight some very recent work published since these two reviews appeared.

### 3.1. SAM-Dependent Methyl Transferases (SAM-Mtases)

Several SAM-Mtases catalyze transfer reactions in which the sulfonium-bound methyl group is replaced with an extended side chain containing a functionality (amine, sulfhydryl, or click chemistry groups) which can be straightforwardly derivatized with commercially available fluorophore-containing reagents. This approach was first used to site-specifically place an AlexaFluor594 dye at the N2 of G26 in an *S. cerevisiae* tRNA^Phe^ transcript by click chemistry, using the enzyme Trm1 that is specific for methylation at this position, and replacing S-Me with S-pent-2-en-4-ynyl [62]. An approach with potential for perhaps broader applicability is the use of SAM-Mtases with strong specificity for sequence contexts within mRNA. In a recent article Ovcharenko et al. [63] demonstrate the potential utility of two mRNA methyltransferases responsible for m^6^A formation, METTL3-14 and METTL16, which target the underlined A in the sequences GGACU and UACAGAGAA, respectively. Replacing the S-Me group in SAM with Se-propargyl affords relatively high yields of N^6^-propargylA (70–80% relative to S-Me) at targeted A residues within model RNAs 20–61 nts long, which are subsequently labeled by click chemistry with Cy5-azide. For METTL3-14, a similar strategy can be used to introduce photocaging groups (e.g., o-nitrobenzyl) at the N^6^ position. An important limitation of this approach is that these two SAM-Mtases can methylate sequences in addition to the most preferred sequences shown above. Thus, for example, METTL3-14 has broad specificity for DRACH sequences (D = A, G, or U; R = G or A; H = A, C or U), making it likely that, in general, several sites in long RNAs containing >150 nts will be targeted by this enzyme.

The enzyme, TrmA, methylates uridine-54 of the T-arm of tRNA, a 17-nucleotide stem loop (nts 49-65) which is structurally conserved and is the minimal substrate for the enzyme [64]. By mutating residues within the TrmA active site and using a high-throughput screen of substrate mutants, Smith et al. [65] identified a mutant TrmA that is unable to methylate a normal substrate derived from *E. coli* tRNA^Phe^, but is a good catalyst for methylating a variant substrate with altered residues in the loop region (nts 54-60). Given the ability of SAM-Mtases to tolerate replacement of the S-Me group with other alkylating agents, as described above, this alteration of specificity might have future utility for introducing fluorophores at selected RNA locations.

### 3.2. tRNA Guanine Transglycosylase (TGT)

Bacterial TGT has also been used to site- specifically label RNAs [66]. As shown in Figure 4, such labeling is achieved by incorporating a 17 nt TGT recognition element into the RNA of interest and using TGT to catalyze the transglycolysis reaction with labeled derivatives of preQ1, its natural substrate. Alkyl linkers have been successfully utilized to append various fluorescent probes to the preQ1 scaffold, including BODIPY, Cy5, Cy7 and thiazole orange.

## 4. Nucleic Acid-Assisted Labeling of Intact RNAs

### 4.1. RNA Acylation at DNA Induced Loops or Gaps

Xiao et al. [67] have introduced an appealingly simple approach which takes advantage of the low reactivity of RNAs in double-stranded DNA-RNA hybrids to use complementary DNA oligonucleotides to protect all but the desired reaction sites in an RNA, by creating loops or gaps as shown in Figure 5. The unprotected RNA nucleotide is then reacted in high yield with the water-soluble nicotinyl acylimidazole (NAI-N_3_), giving an acylated product suitable for further reaction via click chemistry with a suitable fluorophore partner. As a demonstration of the utility of the approach, a 65 nt small nucleolar RNA, SNORD78, was labeled successively with fluorophores at two different positions, Alexa488 at G14 and TAMRA at A49, generating a strong FRET signal. In this approach the sizes of the gaps can be varied, but the most site-specific modifications occur with a loop or gap size of one. Even here, however, small amounts of secondary acylation (20–30%) are found at the position adjacent to the targeted site. Further development of this approach will be needed to suppress such secondary acylation. 

### 4.2. DNA Reactive Sequence Targeting of an Interior Adenosine

Zhao et al. [68,69] have developed a novel approach to specifically transfer an alkyne group to a specific adenosine residue, even one present in a double-stranded region, in a four step sequence. As illustrated in Figure 6, complementary DNA Helper sequences first unfold secondary structure neighboring the targeted A (step i). This is followed by hybridization with a complementary DNA Reactive Sequence derivatized with a precursor group at its 3′-terminus (step ii) and oxidative activation by periodate (step iii), permitting formation of a propargyl N^1^, N^6^–ethenoadenosine (step iv) which can be labeled by click chemistry with a fluorophor. Using this approach, two different A residues targeted within a 275 nt long riboswitch were labeled with yields of each dye derivatized site of between 14–17%. These values are markedly lower than an average value of ~65% which had been previously found for site-specific labeling of single-stranded RNA regions by the same methodology [70], an indication of the inherent difficulty of labeling double- stranded regions. Of note, the periodate oxidation step iii also oxidizes the RNA 3′-terminus to a dialdehyde, permitting its coupling with a second fluor by an orthogonal chemistry. This double labeling approach was used to generate FRET signals from labeled A residues targeted in both single-stranded [71] and double-stranded regions [68,69].

### 4.3. Evolving Ribozymes

Ghaem Maghami et al. [72] have recently demonstrated that the classic Selex approach of in vitro selection of random RNA libraries [73,74] can be used to evolve ribozymes capable of fluorescent labeling of specific interior RNA sites. The basic logic of this approach is illustrated in Figure 7. Once the 40 nt catalytic loop is optimized for labeling of a model sequence in *cis*, the recognition sequences can be altered to target any internal RNA nucleotide in *trans*. The attached reactive groups, amine, ethyne, or azide can be readily derivatized with fluorophores. This procedure has been applied to simple RNA sequences, giving yields of incorporated fluorophores of 50–80%. Substitution of tenofovir diphosphate in place of ATP resulted in formation of a more stable reaction product [75]. More complex RNAs, such as bacterial 5S rRNA and 23S rRNA, have been labeled site-specifically, although the yields of the labeling reactions were not indicated. Subsequent publications will presumably provide additional information for applications of this very promising approach.

## 5. Site-Specific Labeling Requiring De Novo RNA Preparation

### 5.1. Position-Selective Labeling of RNA (PLOR)

PLOR incorporates labels at specific positions of RNAs by combining solid-phase chemical synthesis with highly processive DNA-dependent RNA polymerases (RNAPs) to generate specifically labeled RNAs [76]. PLOR proceeds in three steps: initiation, elongation and termination. In the initiation stage, solid-phase DNA templates are mixed with RNAP, followed by initiation of transcription with the addition of three or fewer types of NTPs, leading to a transcription pause where the missing NTP type is required. Following removal of the residual initial NTPs by solid-phase extraction, an elongation cycle is begun by adding a new NTP mixture, again with three or fewer types of NTPs to resume transcription, which continues until the next point where the cognate NTP is absent. Each pause/resumption step (i.e., each elongation cycle) is controlled by the composition of the added NTPs. Various NTP combinations in each cycle allow incorporation of nucleotides with desired probes into specific positions of the transcript. In principle, this procedure allows modified or labeled NTPs to be introduced at multiple specific positions in nascent transcripts. Once the desired labeling is achieved, all four types of NTPs are added in the termination stage to complete the transcription. The DNA-bearing beads can be recycled after thorough rinsing. 

In one application PLOR has been used to incorporate Cy3 and Cy5 into positions 24 and 55, respectively, of a 71nt adenine aptamer, permitting monitoring of the increase in FRET efficiency on adenine addition. The PLOR approach continues to be developed, most recently by optimizing linker sizes of DNA templates, the lengths and base composition of the RNA synthesized in the initiation stage, and methods for coupling fluorophores to modified nucleotides [77,78].

### 5.2. ^th^G-Containing RNAs

Li et al. [70] have described the assembly of singly modified RNA constructs in which specific G residues are replaced with ^th^G, shown in Figure 8, a highly fluorescent isomorphic guanosine analogue that preserves Watson-Crick base pairing and is available from TriLink Biotechnologies. The method relies on transcription in the presence of excess ^th^G and native nucleoside triphosphates, which enforces initiation with the unnatural analogue. The resulting 5′-end modified transcripts are then mono-phosphorylated and ligated to provide longer site-specifically modified RNA constructs (Figure 8). This approach was used to replace G by ^th^G, one at a time, at four positions within the active center loop region of a hammerhead (HH) ribozyme, with two of the four yielding active ribozymes, and with overall ligation yields of 20–40%. It will be interesting to see whether this method can be applied to other substituted guanosines, or to fluorescent analogues of other nucleosides.

### 5.3. Unnatural Base Pairs

The expansion of the genetic alphabet via the creation of a wide variety unnatural base pairs (UBPs) has been the object of a number of laboratories in recent years, as recently reviewed [79,80]. Site-specific labeling of RNAs is one application of these efforts, as illustrated in Figure 9. Some UBPs innately exhibit strong fluorescence and can used directly in experiments requiring a fluorescent probe [81,82]. The more general approach is to conjugate the X-base with a fluorophore. One example of this approach is provided by the work of Someya et al. [83] who report the fluorescent labeling of RNAs ranging in size from 17–260 nucleotides with high yields and specificity with the derivatized UBP shown in Figure 9B and subsequent click chemistry. Despite the elegance of this approach, it has not as yet been widely applied for RNA labeling because of the lack of commercially available unnatural RNA bases. 

## 6. Conclusions

As this review makes clear, there are many potential methods available for site-specific fluorescent labeling of RNA interior positions, so that the choice of which method to employ will typically depend on the size, sequence and structure of the targeted RNA. That said, it is worth considering whether any of the methods considered could become general, in the sense that it could be readily employed by most laboratories to any RNA of choice. At present, three methods come closest to this goal. Two of these, which label intact RNAs, are RNA acylation at DNA-induced loops or gaps (Section 4.1) and evolving ribozymes (Section 4.3). Each needs further optimization and a clearer demonstration of how well it performs with respect to yield and/or specificity on long, complex RNAs. Such developments should be forthcoming, given the underlying logic of these methods, and the fact that they have only been introduced within the last 1–2 years. In addition, the pause-and-restart PLOR method (Section 5.1) could well become the method of choice for preparing short (<100 nt) labeled RNAs, although its general usefulness for longer RNAs remains to be demonstrated. Two other methods, DNA reactive sequence targeting (Section 4.2), and unnatural base-pairing (Section 5.3), are technically quite impressive. However, each requires reactants which require some expertise in synthetic organic chemistry and are at present not commercially available, limiting their general adoption. The ^th^G method (Section 5.2) has the enormous advantage of incorporating fluorescence with minimal perturbation of RNA structure, and likely can be applied to other fluorescent nucleosides. However, the dependence of this method on RNA ligation, a reaction with quite variable efficiency, could limit its generality. The other methods discussed in this review (Section 2.2, Section 2.3, Section 2.4, Section 3.1 and Section 3.2) each employ enzyme modification on intact RNAs to create sites for fluorescent labeling, with specific sequence and/or structure requirements for such modification to occur. Directed evolution of the modification enzymes could enlarge the pool of RNAs that could be site-specifically labeled by these methods, although it is unlikely that any of them would ever become general.

## Figures and Tables

**Figure 1 molecules-26-01341-f001:**
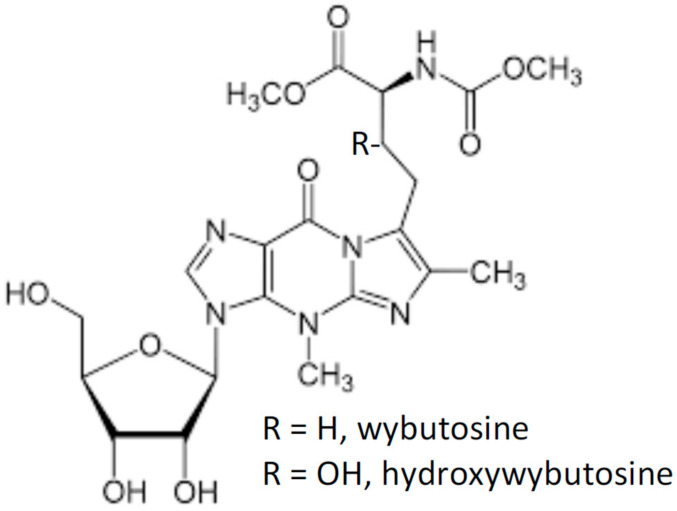
Wybutosines, guanosine derivatives.

**Figure 2 molecules-26-01341-f002:**
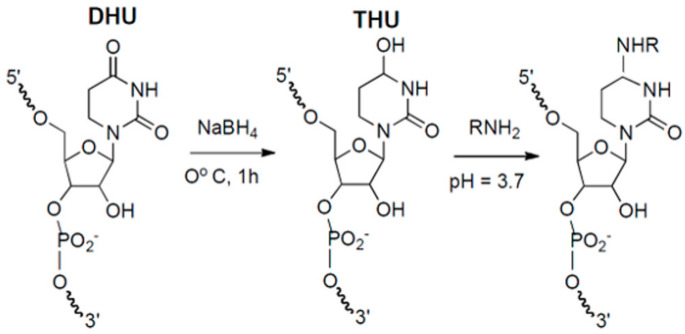
Covalent modification of DHU via THU formation.

**Figure 3 molecules-26-01341-f003:**
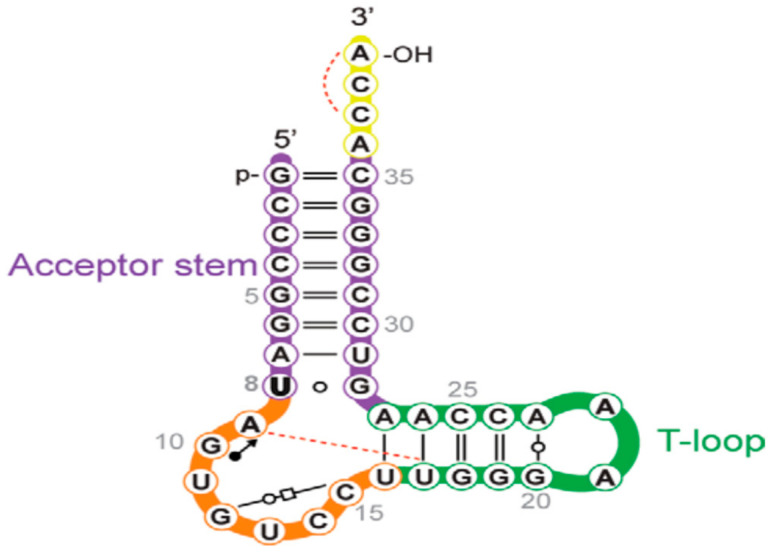
Current minimal substrate for ThiI. Reprinted with permission from reference [58].

**Figure 4 molecules-26-01341-f004:**
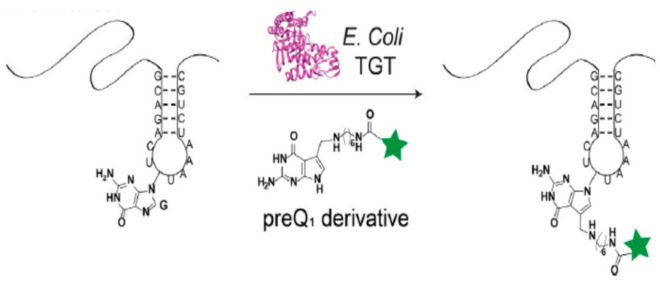
Site- specific TGT labeling of RNA. Reprinted with permission from reference [66].

**Figure 5 molecules-26-01341-f005:**
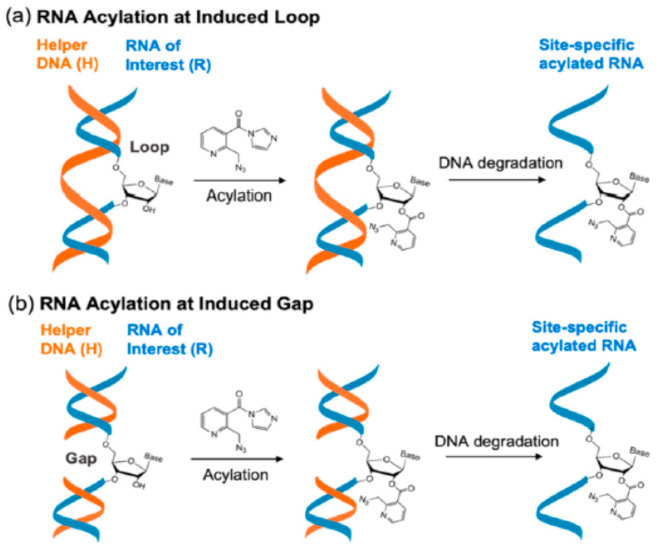
Site-specific modification of RNA loops and gaps induced by DNA. Reprinted with permission from reference [67].

**Figure 6 molecules-26-01341-f006:**
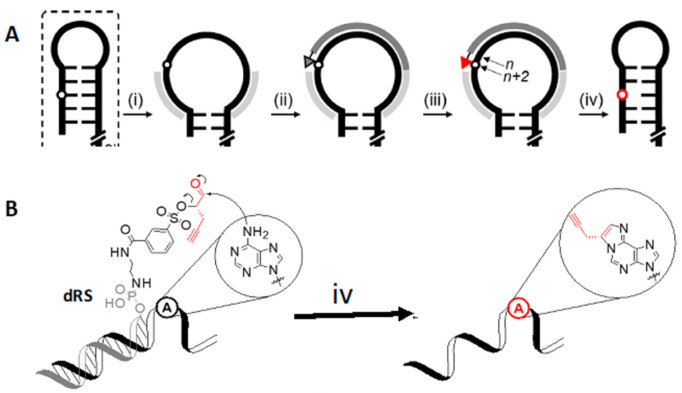
(**A**). (i) DNA Helper Sequences (light gray) annealing, (ii) hybridization of DNA Reactive Sequence (dRS, dark grey) precursor, (iii) activation by sodium periodate, (iv) transfer of the reactive group and modification of an internal adenine. (**B**). Mechanism of reactive group transfer. Adapted with permission from Reference [68].

**Figure 7 molecules-26-01341-f007:**
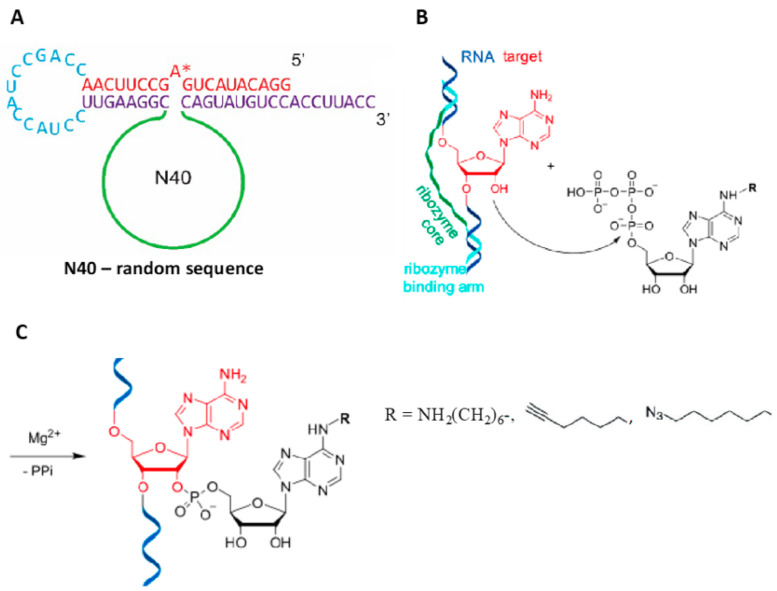
(**A**). RNA catalysts for labeling of a specific adenosine (starred) within a model sequence in cis were identified from a partially structured random RNA pool (N40). The constant region of the RNA pool contained the hypothetical substrate sequence and a pair of recognition arms complementary to the substrate sequence. (**B**). The labeling reaction involving N^6^-substituted ATP adenylylating the 2’ hydroxyl of the targeted adenosine. (**C**). The product of reaction shown in ***B*** showing substitutions used for fluorophore labeling. Adapted with permission from reference [72].

**Figure 8 molecules-26-01341-f008:**
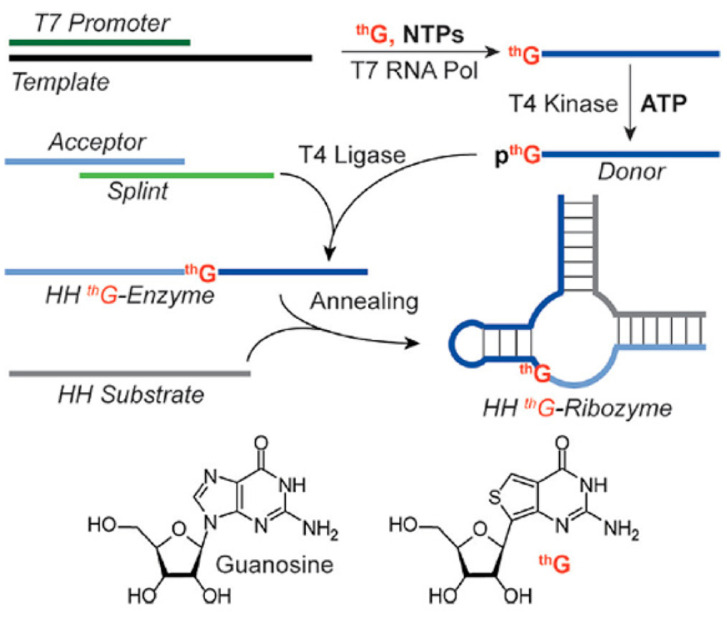
Formation of RNAs site-specifically labeled with ^th^G. Reprinted with permission from reference [70].

**Figure 9 molecules-26-01341-f009:**
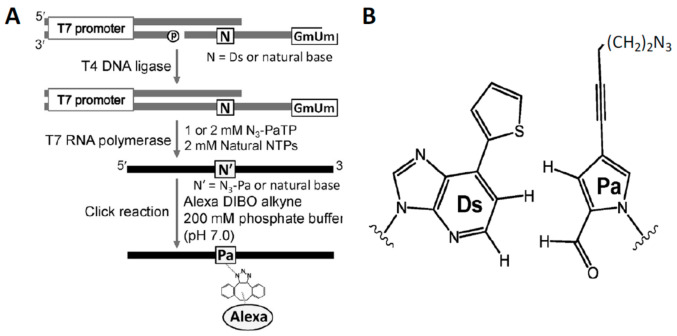
(**A**). The scheme for site-specific incorporation of the unnatural base pair X(Pa)-Y(Ds) into first DNA and then RNA. (**B**). An azide derivative of Pa was used to site-specifically incorporate an Alexa dibenzocyclooctyne fluorophor into several different RNAs by click chemistry. Adapted with permission from reference [83].

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
