# Peer review of "Site-Specific Fluorescent Labeling of RNA Interior Positions"

_molecules, 2021, doi:10.3390/molecules26051341_

Round 1
Reviewer 1 Report
See the attached PDF file.

Author Response
I thank the reviewers for their favorable assessments of the manuscript.
I have made all the changes suggested by reviewer 1 except for one, and I thank the reviewer for so carefully reading the manuscript. The changes are in the attached revision as indicated:
-line 53; lines 89, 90; line 102; line 148; line 151; line 168; line 180; lines 257, 258, 259
I have not made the change suggested for line 109, because the reviewer's objection wasn't clear to me.
In addition, I have made the following minor changes:
-lines 30, 31; line 102
Reviewer 2 Report
This is a useful review of a timely topic. It should be published in Molecules subject to minor revision concerning adding some missing discussion and references. In addition to the subchapters covered, the author should mention the recent works from the Srivatsan lab on
a) terminal uridinyl transferase:
https://doi.org/10.1021/jacs.0c06541
https://doi.org/10.1039/D0CC05092J
b) T7 polymerase incorporation of modified NTPs + posttranscriptional modifications
https://doi.org/10.1093/nar/gky185
https://doi.org/10.1021/acs.bioconjchem.7b00169
https://doi.org/10.1093/nar/gkv903
and a systematic study of the substrate tolerance of T7 RNA polymerase with modified NTPs:
http://dx.doi.org/10.1039/C8OB01498A
Author Response
I thank the reviewers for their favorable assessments of the manuscript.
I have not included direct references to any of the six articles suggested by reviewer 2. Although each of these articles are worthy publications, none are directly pertinent to the main thrust of the manuscript, which is articulated in the title. Moreover, five of the six are cited in the excellent review article co-authored by Professor Srivatsan, in reference (9) of the manuscript, to which the reader is directed for more information on “the introduction of fluorescent probes into RNA by chemical and/or enzymatic procedures”. The most recent article https://doi.org/10.1039/D0CC05092J is not cited in reference (9), but describes a labeling procedure at the 3’-terminus of RNA, not at an interior position, and so is again not pertinent.
Reviewer 3 Report
Comments on the manuscript “Site-specific fluorescent labeling of RNA interior positions”.
The author presents a review on the site-specific fluorescent labeling of RNA in interior positions. Different methods reported in the literature are described by the author with regard to the specific techniques and limitations. The methods described herein are useful to analyze conformational changes of RNA and to detect interactions of RNA with specific binding molecules. The manuscript is well written and easy to follow. It is of interest for scientists working in the field of nucleic acid labelling strategies.
However, introduction and conclusion should be improved. The introduction is very short and provides almost no background information. The reader should be introduced into the field more deeply.
The conclusion appears to consists of several summaries of different chapters. This is rather irritating for the reader and lacks a “take-home message”. I suggest to introduce a short summarizing section at the end of each chapter and to write a shorter conclusion including a “take-home message” for the reader.
After having made these changes, I recommend this review for publication in “Molecules”.
Author Response
I thank the reviewers for their favorable assessments of the manuscript.
I disagree with the editorial judgment of the reviewer 3. This is a short article, and in my judgment is amenable to a summary at the end, rather than after each short section.